# Near-Field High-Resolution SAR Imaging with Sparse Sampling Interval

**DOI:** 10.3390/s22155548

**Published:** 2022-07-25

**Authors:** Chengyi Zhao, Leijun Xu, Xue Bai, Jianfeng Chen

**Affiliations:** School of Electrical and Information Engineering, Jiangsu University, Zhenjiang 212000, China; 2222007065@stmail.ujs.edu.cn (C.Z.); baixue@ujs.edu.cn (X.B.); 1000005767@ujs.edu.cn (J.C.)

**Keywords:** millimeter-wave, near-field synthetic aperture radar (SAR), frequency-modulated continuous-wave (FMCW), sparse sampling image reconstruction

## Abstract

Near-field high-resolution synthetic aperture radar (SAR) imaging is mostly accompanied by a large number of data acquisition processes, which increases the system complexity and device cost. According to extensive reports, reducing the number of sampling points of a radar in space can greatly reduce the amount of data. However, when spatial sparse sampling is carried out, a ghost normally appears in the imaging results due to the high side lobes generated in the azimuth. To address this issue, a technique is introduced in this paper to recover the blank data through amplitude and phase compensation based on the correlation between sparse array sampling through adjacent points. Firstly, the data sampled by the sparse array is compressed in the range direction to obtain the expected data slices in the same range direction. Then, the blank element of the slice is compensated for with amplitude and phase to obtain full aperture data. Finally, the matched filter method is used to aid in the image reconstruction. The simulation results verified that the method proposed in this paper can effectively reconstruct the image under two kinds of sparse sampling conditions. Thus, a simple single-input single-output (SISO) synthetic aperture radar imaging test bench is established. Compared with the results of a 1 mm (1/4 λ) sampling interval, the quality of the reconstructed image under the condition of a 4 mm (1 λ) sampling interval still stands using our proposed method. Demonstrated by the experiment, the normalized root-mean-square error(NMSE) is 5.75%. Additionally, when the spatial sampling points are sampled randomly with 30% of the full sampling condition, this method can also restore and reconstruct the image with high quality. Due to the decrease of sampling points, the data volume can be reduced, which is beneficial for improving the scanning speed and alleviating the pressure of data transmission for near-field high resolution SAR imaging systems.

## 1. Introductions

Near-field millimeter-wave (mmWave) imaging technology is gaining grounds in recent years and now plays a significant role in many spheres, including secure checks, medical care and nondestructive detection [1,2,3,4,5]. Due to the characteristics of non-invasive and non-destructive testing, the demand for millimeter-wave imaging systems is further increasing. For this reason, a considerable amount of research has been done to improve the scanning structure and reconstruction algorithm.

For near-field imaging, the plane wave assumption is invalid, and therefore, the curvature of the wavefront must be fully compensated in the process of image reconstruction. The back projection algorithm (BPA) [6,7] provides a straightforward method to reconstruct the target area by correlating reflection characteristics of each spatial coordinate, but this method also suffers from a large amount of calculation. The synthetic aperture radar (SAR) imaging algorithm, which is based on the fast Fourier including range migration algorithm (RMA) [8,9] (also named the ω-K algorithm [10]) and the match filter algorithm [11], provides an avenue to improve the speed of reconstruction. However, the practical speed of the general SAR imaging algorithm is limited by the amount of required data, which results from the demand for high resolution and the sampling condition of Nyquist criteria. Generally, the scanning points that a high-resolution SAR system need are often as large as tens of thousands.

To alleviate the above problem, much effort has been devoted both at the software and hardware levels. For instance, at the hardware level, multiple-input multiple-output (MIMO) technology is widely used in radar imaging [12,13,14]. With MIMO technology, for an imaging system with N transmitters and M receivers, the points that the radar needs to traverse in space are reduced to the original 1/(N × M). According to [15], the fast reconstruction of multi-static SAR data is realized by using Fourier-based reconstruction along a uniform direction in *k*-space and reconstructing nonuniform directions via SAR-based methods. Ref. [16] proposed a method to calibrate MIMO array data, which simplified multi-static data to mono-static data for processing. The image was reconstructed by fast Fourier transform, which demonstrated the high-precision reconstruction of sparse interval sampling.

At the software level, compressive sensing (CS) [17,18] and matrix completion (MC) [19,20,21] methods have been proposed and applied in some SAR applications, which allows reconstructing an image from sparse or compressible signals. The CS method helps to avoid the direct sampling of the original signal at the receiving end and adopts a novel algorithm to recover the compressible signal, which greatly simplifies the radar structure. For the matrix completion (MC) method, a low-rank matrix from a subset of matrix entries can easily be recovered. In [22], the CS-based method was proposed for image reconstruction through l1-norm minimization to overcome the non-uniform data. In [23], MC theory was applied in the process of SAR imaging. Through the low-rank characteristics of a radar echo data matrix, the fully aperture data were reconstructed from under-sampled data.

For some time now, how to reduce the amount of data while, at the same time, maintaining the imaging quality has been quite challenging. MIMO technology reduces the spatial sampling points that the radar needs to traverse in the scanning process. However, due to its multi-channel characteristics, it is necessary to calibrate the phase of echoes from different channels [24]. Meanwhile, the data matrix at the input of the algorithm has not been reduced, hence the system hardware structure becomes more complex. CS makes sparse sampling reconstruction possible, but it makes use of the sparse characteristics of the signal; therefore, there is the need to design a complex observation matrix. The reconstruction quality is also limited by the performance of the observation matrix.

In this paper, an effective near-field SAR imaging method with a large sampling interval or a spatial under-sampling condition is proposed. Different from the CS method, our method uses the range focusing data as the basis of reconstruction and does not need to design an extra observation matrix. The full aperture sampling data matrix is then reconstructed from the under-sampling data, which is consistent with the Nyquist theorem. Match filter processing is then employed for the image reconstruction. Numerical simulation and experiment validate that, in case of spatial under sampling, the near-field SAR image can be effectively reconstructed with considerable quality.

The rest of this paper is organized as follows: Section 2 reviews the match filter imaging method based on the frequency modulated continuous wave (FMCW) signal and the imaging algorithm based on amplitude and phase compensation recovery. Section 3 presents the experimental methods and results, which are followed by the conclusion in Section 4.

## 2. Module and Algorithm

### 2.1. Near-Field SAR Module

Suppose the radar sensor is at xoy plane (described as the scanning plane), where z=0, while the target is placed at the plane where $z=z_{0}$.

$D_{x}$ and $D_{y}$ are the scanning ranges of an antenna in the horizontal and vertical directions, respectively, which encircle the area of the synthetic aperture. The synthetic aperture is shown in Figure 1. In the established Cartesian coordinate system, the *x* axis, *y* axis and *z* axis indicate the horizontal, vertical and range directions, respectively. The xoy plane donates the radar scanning plane, whiles the target is placed at $\left(x,y,z_{0}\right)$, and the reflection of target is considered as f(x,y). In a two-dimensional scanned near-field SAR, the echo signal can be expressed as follows:(1)$$s\left(x^{{}^{\prime }},y^{{}^{\prime }}\right)=\;\int \int \;f\left(x,y\right)\times exp\left(-jk\sqrt{{\left(x-x^{{}^{\prime }}\right)}^{2}+{\left(y-y^{{}^{\prime }}\right)}^{2}+z_{0}^{2}}\right)dxdy$$

For near-field SAR, spherical waves can be decomposed into the superposition of countless plane waves as shown in (2), and substituted into Equation (1) to obtain Equation (3) given by:(2)$$exp\left(-jk\sqrt{{\left(x-x^{{}^{\prime }}\right)}^{2}+{\left(y-y^{{}^{\prime }}\right)}^{2}+z_{0}^{2}}\right)\approx exp\left(-jk_{x}\left(x-x^{{}^{\prime }}\right)-jk_{y}\left(y-y^{{}^{\prime }}\right)+k_{z}z_{0}\right)$$
(3)$$s\left(x^{{}^{\prime }},y^{{}^{\prime }}\right)=\;\int \int \;\left[\;\int \int \;f\left(x,y\right)\times exp\left(-j\left(k_{x}x+k_{y}y\right)\right)dxdy\right]\times exp\left(j\left(k_{x}x^{{}^{\prime }}+k_{y}y^{{}^{\prime }}+k_{z}z_{0}\right)\right)dk_{x}dk_{y}$$

Applying two-dimensional discrete Fourier transform equation, $s(x^{{}^{\prime }},y^{{}^{\prime }})$ can be expressed as:(4)$$s\left(x^{{}^{\prime }},y^{{}^{\prime }}\right)=\;\int \int \;FFT_{2D}\left(f\left(x,y\right)\right)\times exp\left(jk_{z}z_{0}\right)\times exp\left(j\left(k_{x}x^{{}^{\prime }}+k_{y}y^{{}^{\prime }}\right)\right)dk_{x}dk_{y}$$

Equation (4) gives:(5)$$FFT_{2D}\left(s\left(x^{{}^{\prime }},y^{{}^{\prime }}\right)\right)exp\left(-jk_{z}z_{0}\right)=FFT_{2D}\left(f\left(x,y\right)\right)$$

According to the dispersion relation for plane waves in free space, it is possible to write:(6)$$k_{x}^{2}+k_{y}^{2}+k_{z}^{2}={\left(2k\right)}^{2}$$

Thus, the target scattering coefficient f(x,y) can be expressed as follows:(7)$$f\left(x,y\right)=FFT_{2D}^{-1}\left(FFT_{2D}\left(s\left(x^{{}^{\prime }},y^{{}^{\prime }}\right)\right)\times exp\left(-jk_{z}z_{0}\right)\right)$$

Figure 2 shows the complete workflow of matched filter image reconstruction. First, the radar carries out two-dimensional scanning in space and collects echoes to obtain the data matrix composed of time-domain data. Then FFT is performed on the time domain to obtain the range compressed data. Then 2DFFT is performed on it to obtain azimuth compressed data. After that, a matched filter for a specific distance is generated to calculate the Hadamard product with the azimuth compressed data. The 2D Fourier inverse transform is performed on the obtained result, and the image of the target area is obtained by taking the modulus display.

### 2.2. Under-Sampling Recovery Reconstruction Method

As shown in Figure 1, when the radar is at a certain point $\left(x_{i},y_{i},0\right)$ in the scanning process, the transmitted signal from the transmitting antenna, TX, can be expressed as:(8)$$m\left(t\right)=cos(2\pi \left(f_{0}t+0.5Kt^{2}\right)),0<t<T$$
where $f_{0}$ is the starting frequency of the FMCW signal, and K=B/T indicates the chirp rate. For ease of understanding, it is assumed that TX and Rx overlap in space. Suppose a target is at (x,y,z), and the echo signal captured by the RX antenna is given as:(9)$$r\left(t-\tau \right)=\beta cos(2\pi \left(f_{0}\left(t-\tau \right)+0.5K{\left(t-\tau \right)}^{2}\right))$$
where τ is the time required for the radar signal to propagate between the radar and target, and β represents the attenuation caused by the target reflectivity and system.
(10)$$\tau =\frac{2r}{c}=\frac{2\sqrt{{\left(x-x_{i}\right)}^{2}+{\left(y-y_{i}\right)}^{2}+z_{0}^{2}}}{c}$$

The signal after dechirp demodulated with the in-phase and quadrature can be expressed as:(11)$$s\left(t\right)=I\left(t\right)-jQ\left(t\right)=\beta e^{-j2\pi (f_{0}\tau +K\tau t-0.5K\tau^{2})}$$
where $0.5K\tau^{2}$ is known as the residual video phase (RVP), which can be ignored in the imaging process. The instantaneous frequency is expressed as $f=f_{0}+Kt$. Thus, the signal will be $s\left(\tau \right)=\beta e^{-j2\pi f\tau }.$

Additionally, suppose the target is located at $\left(x,y,z_{0}\right)$, and the radar acquires echo signal $s\left(\tau_{1}\right)=\beta e^{-j2\pi f\tau_{1}}$ at $\left(x_{1},y_{1},0\right)$; then, the radar moves to location $\left(x_{n},y_{i},0\right)$. The echo signal can also be expressed as $s\left(\tau_{n}\right)=\beta e^{-j2\pi f(\tau_{1}+\Delta \tau_{i,n-1})}$.

$\Delta \tau_{i,n-1}$ represents the change in the echo delay caused by the change in the radar position; therefore, the time domain echo signal matrix can be expressed as follows:(12)$$s_{time}=\left(\begin{array}{ccccc} s(\tau_{1}) & s(\tau_{1}+\Delta \tau_{1,1}) & s(\tau_{1}+\Delta \tau_{1,2}) & \cdots & s(\tau_{1}+\Delta \tau_{1,n-1})\\ s(\tau_{2}) & s(\tau_{1}+\Delta \tau_{2,1}) & s(\tau_{1}+\Delta \tau_{2,2}) & \cdots & s(\tau_{1}+\Delta \tau_{2,n-1})\\ \vdots & \vdots & \vdots & \ddots & \vdots \\ s(\tau_{m}) & s(\tau_{1}+\Delta \tau_{m,1}) & s(\tau_{1}+\Delta \tau_{m,2}) & \cdots & s(\tau_{1}+\Delta \tau_{m,n-1}) \end{array}\right)$$

In the MF reconstruction algorithm, the time domain signal obtained by radar needs to be converted into range-focused data by fast Fourier transform (FFT). The radar echo data will form slices after range focusing as shown in Figure 3. Each slice corresponds to a matrix in the range gate that can be used to reconstruct the image.

During the modeling of the near-field SAR reconstruction, the radar is in a “stop-go-stop” working mode. The range migration caused by the change in the imaging distance, target position and radar position will not exceed the range of the same range gate. In other words, the information of the imaging target basically falls in the same distance as the focusing slice. The matrix above shows that there is a strong correlation between various elements. Additionally, once the FFT is performed, it is expected that there will be another strong correlation. Therefore, when there is an element vacancy, it can be replaced by calculating a reasonable value from nearby elements.

On a data slice focused in the same range gate, range-focusing data of a target received by the radar at different positions will vary in amplitude and phase. We can then represent the response of a target on a certain distance slice as (ρ(r),ϕ(r)) in polar coordinates, where ρ(r) and ϕ(r) denote the amplitude and the phase, respectively.

During the process of recovering missing elements, the change in amplitude of similar sampling points is usually mild. Moreover, this amplitude is inversely proportional to the distance; thus, the blank on the amplitude can be recovered by uncomplicated interpolation methods. For radar ranging, the phase change caused by subtle range change can be described as Δϕ=4πΔr/λ. This can be used to calculate blank elements.

In the practical application of near-field SAR, it is recommended that the distance between the apertures be less than λ/4. Therefore, during the data recovery process, the radar sensor must be placed at the plane z=0 (described as the scanning plane) to restore the data matrix in order to achieve the desired interval.

As shown in Figure 4, when $\Delta x<<z_{0}$, then R1−R2 can be expressed as follows:(13)$$R1-R2=\left(x_{B}-x_{peak}\right)\times \Delta x/\sqrt{\left({\left(x_{B}-x_{peak}\right)}^{2}+z_{0}^{2}\right)}$$

In (13), $x_{peak}$ is the approximate target position in this direction (and can be obtained from the amplitude information), and $x_{B}$ is the location of the element to be calculated.

With the above analysis and detailed studies, the near-field image reconstruction method is now summarized in the next algorithm table. Figure 5 shows the flow chart of the proposed method. Algorithm 1 shows the detailed process.
**Algorithm 1** Spatial under-sampling image reconstruction method based on amplitude and phase compensation**Input:** under-sampled data with coordinate markers $s(x_{s},y_{s},t)$**Output:** Image reconstruction results   Initialize all parameters of synthetic aperture radar   The time domain data of each coordinate is compressed $S(x_{s},y_{s},Z_{range})$   Obtain the distance focus data slice of the estimated distance $S\left(x_{s},y_{s}\right)=\left(\rho \left(x_{s},y_{s}\right),\phi \left(x_{s},y_{s}\right)\right)$   Obtain the full amplitude matrix $\rho (x_{full},y_{full})$ through $\rho (x_{s},y_{s})$ with interpolation   Construct the phase data matrix to be recovered $\phi (x_{full},y_{full})$ with $\phi (x_{s},y_{s})$
**for all**
$\left(x_{i},y_{i}\right)\in \left(x_{full},y_{full}\right)$
**do**     **if**
$\rho (x_{i},y_{i})=0\;or\;Null$
**then**       replace $\rho (x_{i},y_{i})$ with calculated results from nearest known element     **end if**   **end for**   $S\left(x_{full},y_{full}\right)=\rho \left(x_{full},y_{full}\right)*\left(cos\left(\phi \left(x_{full},y_{full}\right)\right)+jsin\left(\phi \left(x_{full},y_{full}\right)\right)\right)$   Create match filter H(x,y)   Reconstruction of target area $f\left(x,y\right)=FFT_{2D}^{-1}\left(FFT_{2D}S\left(x_{full},y_{full}\right)\times FFT_{2D}H\left(x,y\right)\right)$

## 3. Experimentation and Results

Throughout the experiments, the following parameters were used: (1) the FMCW radar operated at 77 GHz with a bandwidth of 4 GHz and a slope K=63.34 MHz/μs, (2) the wavelength λ was computed as 3.89 mm (wavelength corresponding to 77 GHz), and (3) the target distance $Z_{0}$ was 300 mm. Furthermore, all the images presented below are reconstructed on a host PC with a Ryzen 5600 × 3.7 GHz central processing unit (CPU) and 16 gigabytes of random-access memory (RAM).

### 3.1. Simulation

Firstly, the feasibility of this method in image reconstruction is verified by simulation. The parameters are shown in Table 1.

Figure 6 shows the phase and amplitude response of one-line aperture data in the simulation of a point target. After pretreatment of the 30% under-sampled data by the method mentioned above, the recovered results are very close to the full sampling data. Table 2 depicts the errors associated with the recovered data, which is obtained from the sampled data with a different under-sampled rate (USR). From the comparison results given in Table 2, it can be seen that the method has advantages in reducing the errors.

In order to verify the applicability of this method in image reconstruction, a target, shown in Figure 7, is simulated. The following two cases are considered.

In the first case, the $d_{x}$ and $d_{y}$ of the synthetic aperture radar is increased. The reconstructed image of original data and the ones by proposed method are shown in Figure 8a,b, where $d_{x}$ = $d_{y}$ = 4 mm.

For the second case, $d_{x}$ and $d_{y}$ of the synthetic aperture radar are kept the same, while 30% of the source data are used for image reconstruction. Figure 8c shows the results of the reconstructed image by the CS method, and Figure 8d shows the results of the method proposed in this paper.

As shown above, we have evaluated the reconstructed images from subjective vision. Then, a quantitative analysis was conducted on the reconstructed images with under-sampling conditions and those under normal sampling conditions using normalized root-mean-square error (*NMSE*), peak signal-to-noise ratio (*PSNR*) and structural similarity (*SSIM*). The image under the conventional sampling condition is represented by *x*, and *y* represents the reconstructed image under the under-sampling condition. *M* and *N* represent the length and width (number of pixels) of the image, respectively.

*NMSE* is widely used to measure the error between the reconstructed image and the fully sampled image, as shown in the equation. The smaller the value of *NMSE*, the closer the reconstructed image is to the full sampled image.
(14)$$NMSE=\frac{||x-{y||}_{2}^{2}}{{\left||y|\right|}_{2}^{2}}$$

*SSIM* can also measure the similarity of two images from amplitude, contrast and structure. It effectively reflects the proximity of image edges, textures and details. Its value is between 0 and 1. The closer it is to 1, the closer the reconstructed image is to the original image and the higher the quality is. In Equation (Equation 15), $\mu_{x}$ and $\mu_{y}$ represent the mean value of the full sampling result *x* and the under-sampling recovery result *y*, respectively. $\sigma_{x}$ and $\sigma_{y}$ are the variances of *x* and *y*, respectively. $\sigma_{xy}$ is the covariance of *x* and *y*.
(15)$$SSIM=\frac{\left(2\mu_{x}\mu_{y}+C_{1}\right)\left(\sigma_{xy}+C_{2}\right)}{\left(\mu_{x}^{2}+\mu_{y}^{2}+C_{1}\right)\left(\sigma_{x}^{2}+\sigma_{y}^{2}+C_{2}\right)}$$

*PSNR* is an objective evaluation index widely used in the field of image processing. It represents the amount of original signal in the reconstructed image, and the larger the value is, the closer the reconstructed image is to the original image.
(16)$$PSNR=10log_{10}\frac{max{\left(y\right)}^{2}}{MSE}$$
(17)$$MSE=\frac{1}{M\times N}||x-{y||}_{2}^{2}$$

Table 3 shows the summary of the target image reconstructed by this method under different sampling conditions and various indicators under full sampling conditions. Large interval sampling means that the radar scanning step increases, which will lead to a decline in imaging accuracy and even the occurrence of ghosting. Random sparse samples means that we randomly select some spatial sampling points from the fully sampled data for sampling in the matched filter method; this will also lead to a reduction in accuracy and an increase in image noise compared with the results of full sampling reconstruction. It is inferred from Table 3 that the image qualities under different sampling conditions can still be guaranteed by using amplitude and the phase compensation reconstruction method.

### 3.2. Near-Field SAR Testbed

Due to the development of system-on-chip mmWave sensor technology, commercial off-the-shelf (COTS) mmWave radar EVM is available. IWR1843Boost EVM developed by Texas Instruments is selected to set up the near-field SAR testbed. The EVM consists of three transmitting antennas and four receiving antennas, which can support up to 4 GHz bandwidth on the 77–81 GHz frequency band. Due to the excellent performance of on-board DSP, the process of range focusing of radar echo data is executed within IWR1843Boost EVM. Compared with the testbeds in [25], this can not only save the cost of high-speed acquisition card, but also simplify the system structure and improve the operation efficiency to a certain extent.

Figure 9 shows that the SAR testbed consists of several major parts: (1) a mmWave sensor; (2) a two-axis mechanical scanner; (3) an Arduino UNO with a motor driver; (4) a UI integrated with the reconstruction algorithm.

Figure 10 illustrates the system architecture of the SAR testbed. Arduino UNO with two motor drivers constitutes the controller of the scanning platform. Through the serial port, range focus data uploaded by the radar in real time can be obtained during the scanning process and synchronized with the current radar coordinates provided by the controller through the interior of the UI. This provides the possibility for real-time SAR data processing in the future.

### 3.3. Measured Data and Reconstruction Results

The method proposed in this paper was tested on the above test-bed. In all experiments, the radar worked in single-input single-output (SISO) mode. A wrench with the length of 125 mm was placed in front of the radar. The SAR aperture was synthesized to cover an area of $D_{x}\approx 160$ mm by $D_{y}\approx 120$ mm.

A. Full sampled condition

The spatial sampling intervals were selected as 0.91 mm and 1 mm, respectively. The platform drove the radar to scan all points on the synthetic aperture and carry out real-time data transmission. The whole process lasted about 5 min. Figure 11 shows the target image within a visible region of 300 × 300 ${\mathrm{mm}}^{2}$. This result was used as an evaluation standard to evaluate the recovery results of subsequent under-sampling conditions.

B. Under-sampled condition

The spatial sampling intervals were changed to $d_{x}=3.84$ mm, $d_{y}=4$ mm, which was four times the intervals above. Figure 12a shows the image reconstructed by MF directly with the original SAR data. The aliasing image caused by the sampling interval not satisfying the Nyquist theorem was located at a distance of (0, ±168 mm), (±156 mm, 0).

The SAR data obtained within large spatial sampling intervals was preprocessed by the method proposed by this article and reconstructed. The image shown in Figure 12b indicates the effectiveness of this method. The ghost image due to aliasing was eliminated. The jagged shape of the target edge also became smooth, and the image details were guaranteed at the same time. Intuitively, the result is in high consistency with the image in Figure 11.

Next, another under sampling condition was tested. The data obtained by small interval sampling above were randomly sampled with different under-sampled rates. Under this condition, in order to verify the recovery effect of this method in different situations, the value range of USR was set at 20–90%. Figure 12c shows an image reconstructed by MF from 60% USR data. As a comparison, Figure 12d shows the target image reconstructed after preprocessing. Figure 12e,f are the results of USR = 30%. Judging from the noise of the whole image and the details of the target edge, the imaging effect of the latter is obviously better than that of the former.

The images reconstructed by our proposed method with different under-sampling conditions were evaluated by comparing with the result reconstructed from full sampled data. Table 4 presents NMSE, SSIM and PSNR of the reconstructed image by the proposed method with different USRs.

## 4. Conclusions

In this paper, a recovery and reconstruction method based on amplitude and phase compensation of adjacent sampling points is proposed. It allows full aperture data to be recovered from spatial under-sampling during SAR scanning and reconstructed by the MF method. This method is mainly based on the following assumptions. Between adjacent sampling points (generally less than λ/4), the echo signal caused by the micro-distance change of has a certain correlation in the range compressed data. Thus, through amplitude and phase compensation, the blank sampling point at a certain position is restored, and the full sampling data is obtained. After this processing, the MF method can reconstruct it and effectively avoid ghosting and image quality degradation caused by undersampling.

The main feature of this method is that it allows the under sampling process in MF reconstruction and will not be affected by the correlation phenomenon caused by the high side lobe of undersampling. The simulation and experimental results clearly show that this method can effectively reconstruct the target in both undersampling cases, and effectively suppress some adverse phenomena, especially ghosting. Unlike the CS method, this method does not need to design an observation matrix. The test platform of this paper also confirms that this method can recover the range-focused data directly (this process can be realized by on-board DSP) without sampling the time-domain data, which greatly reduces the cost of data transmission.

The amplitude and phase compensation method proposed in this paper can recover the sparse sampled data, and the target image with good performance can be reconstructed by MF method. This method assumes that the detection scene is a continuous curve on the range compressed data of radar echo, and the SAR imaging process is equivalent to sampling and restoring this curve. In future work, we will try from two aspects. Firstly, in terms of computational performance, the current method in this paper needs to traverse all points in the data recovery process of sparse sampling, which has a lot of optimization space. Secondly, in the imaging model, only the possibility of two-dimensional imaging is considered at present. Introducing appropriate distance compensation and extending this method to three-dimensional is also a focus in the future.

## Figures and Tables

**Figure 1 sensors-22-05548-f001:**
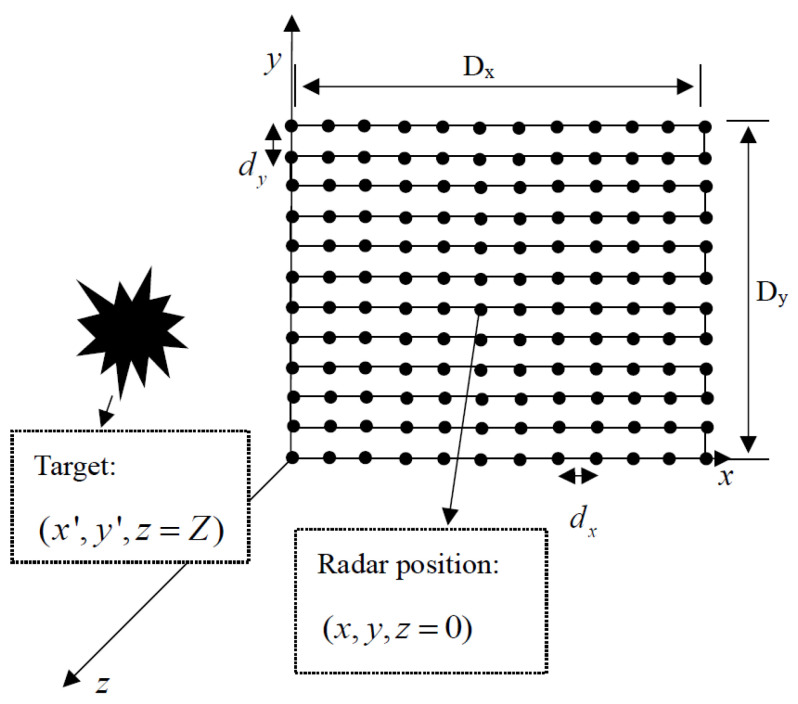
Geometric diagram of two-dimensional scanning synthetic aperture radar in the near-field environment.

**Figure 2 sensors-22-05548-f002:**
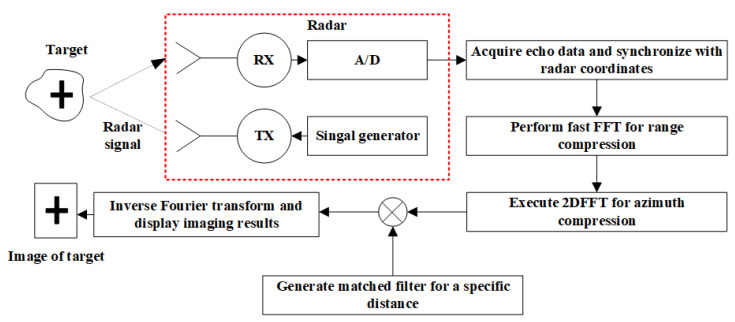
Flowchart of matched filter reconstruction for near-field SAR image; The red box shows the schematic diagram of the radar module.

**Figure 3 sensors-22-05548-f003:**
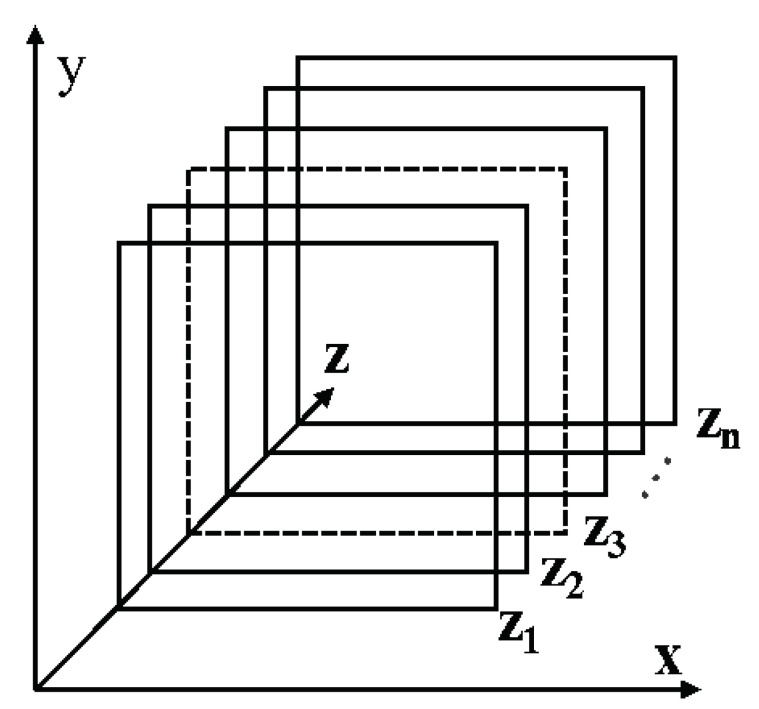
Range compressed data after 1DFFT is performed on the time axis of echo data.

**Figure 4 sensors-22-05548-f004:**
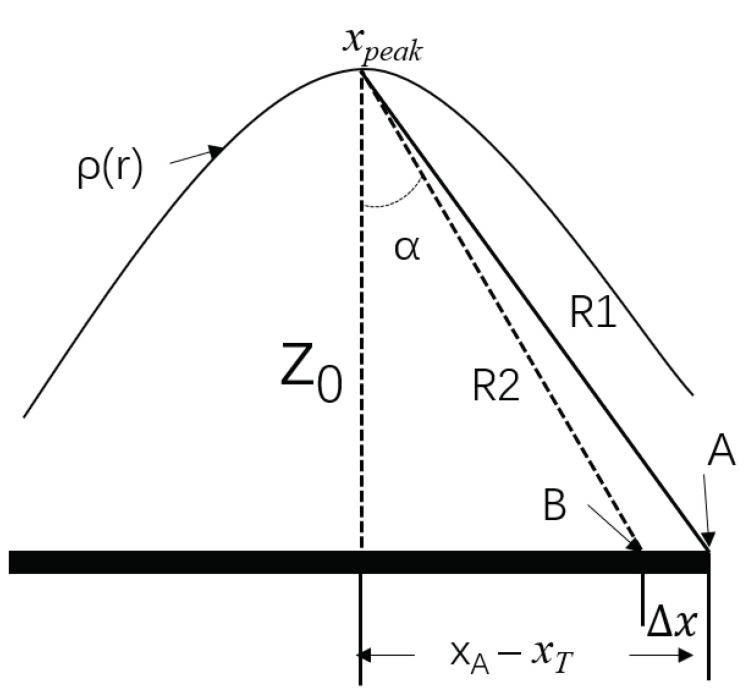
Diagram of a row of distance compressed data in the *x* direction.

**Figure 5 sensors-22-05548-f005:**
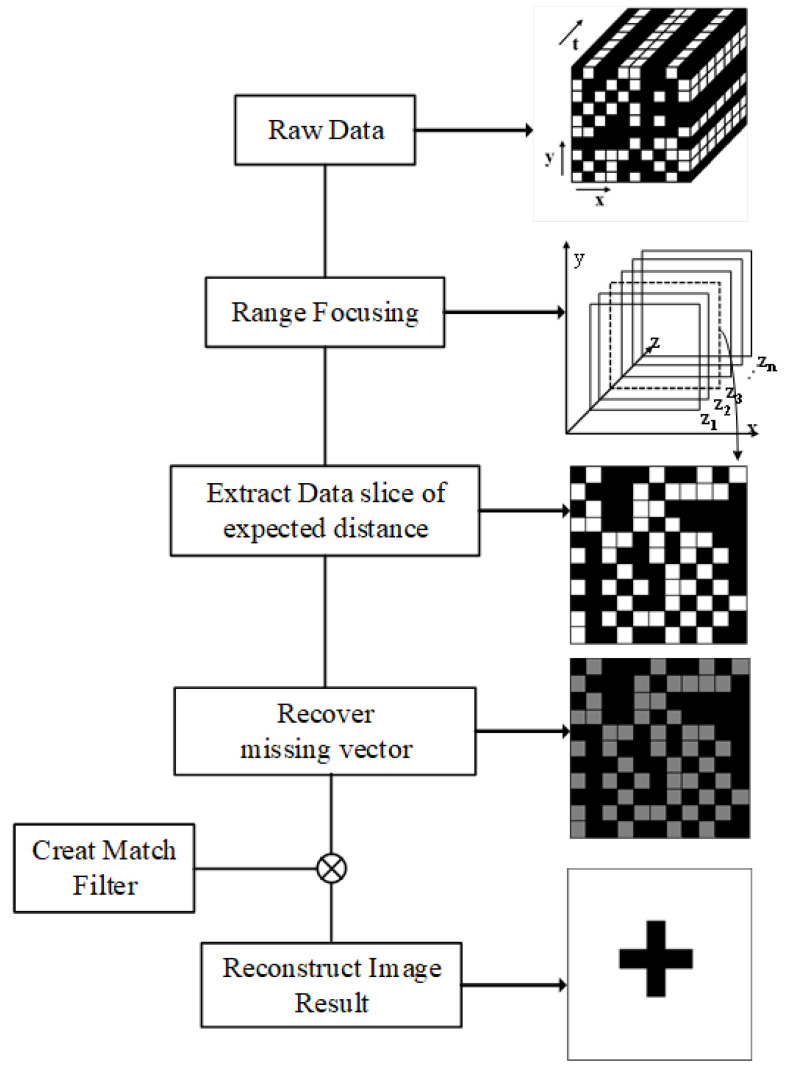
Flow diagram of spatial under-sampling image reconstruction method based on amplitude and phase compensation.

**Figure 6 sensors-22-05548-f006:**
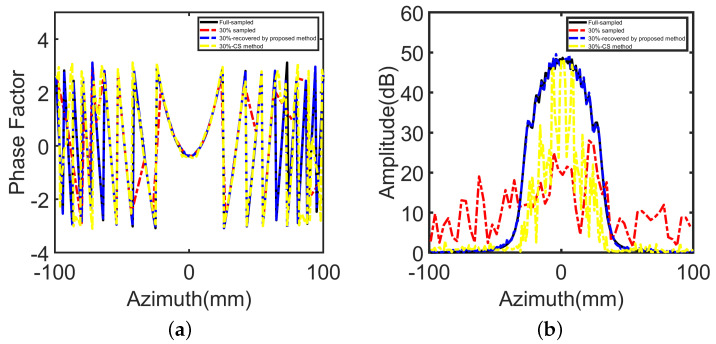
Phase factor and azimuth FFT result of a single line in the range-focusing data slice: (**a**) phase factor, (**b**) azimuth FFT result.

**Figure 7 sensors-22-05548-f007:**
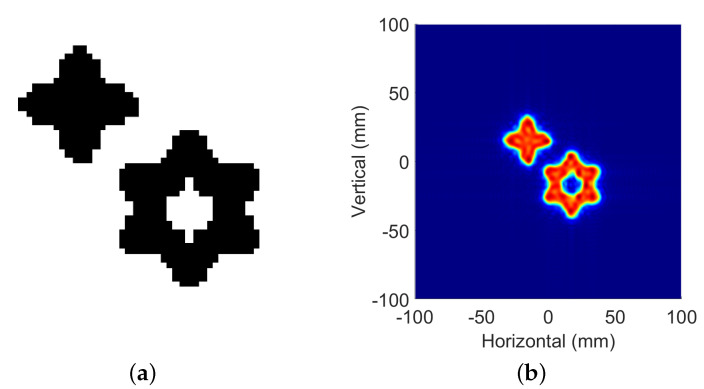
Simulated target and reconstructed image from full-sampled data: (**a**) simulation target; (**b**) reconstruction result.

**Figure 8 sensors-22-05548-f008:**
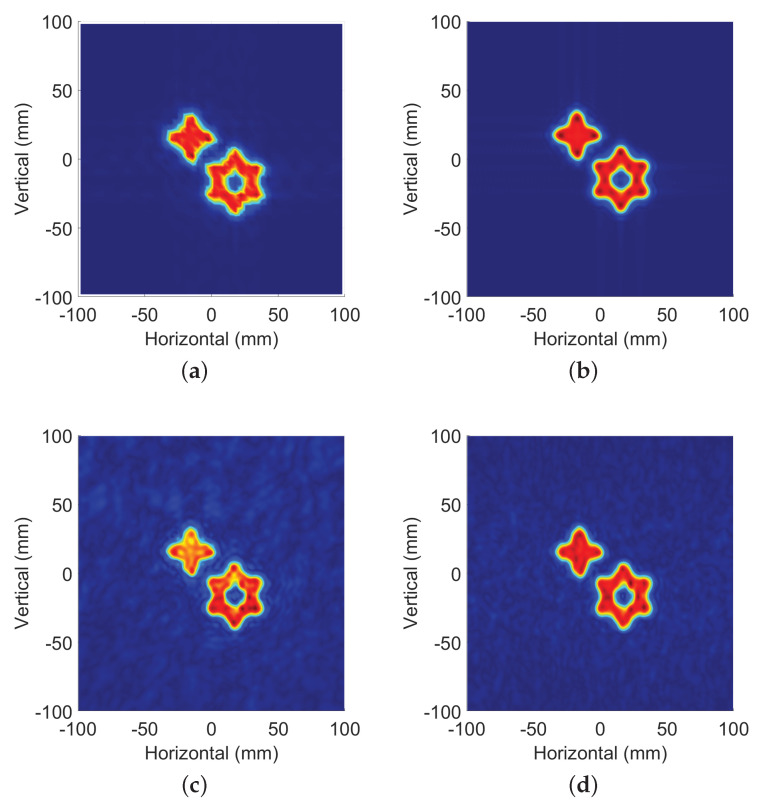
Reconstruction results under different conditions: (**a**) reconstruction results of the SAR data where $d_{x}$ = $d_{y}$ = 4 mm without preprocess; (**b**) reconstruction results of the SAR data where $d_{x}$ = $d_{y}$ = 4 mm with preprocessing proposed by this article; (**c**) reconstruction results of the SAR data with USR = 30% by CS method; (**d**) reconstruction results by the method proposed by this article where the USR of data is 30%.

**Figure 9 sensors-22-05548-f009:**
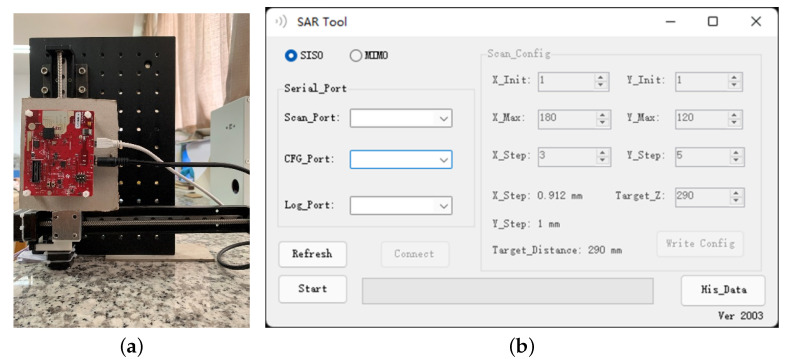
Near-field SAR testbed: (**a**) hardware structure; (**b**) software UI.

**Figure 10 sensors-22-05548-f010:**
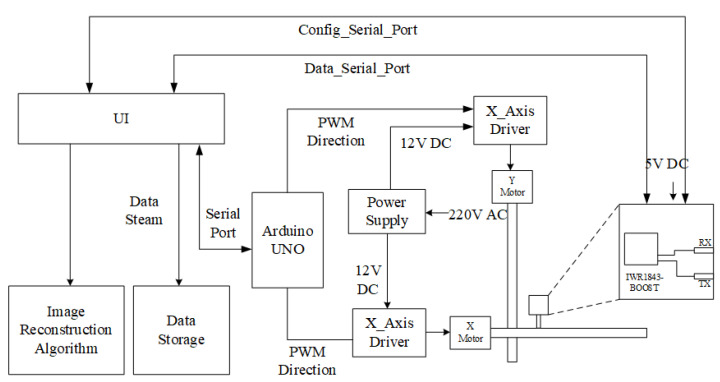
The architecture of the SAR testbed consists of a mmWave sensor and a mechanical scanner.

**Figure 11 sensors-22-05548-f011:**
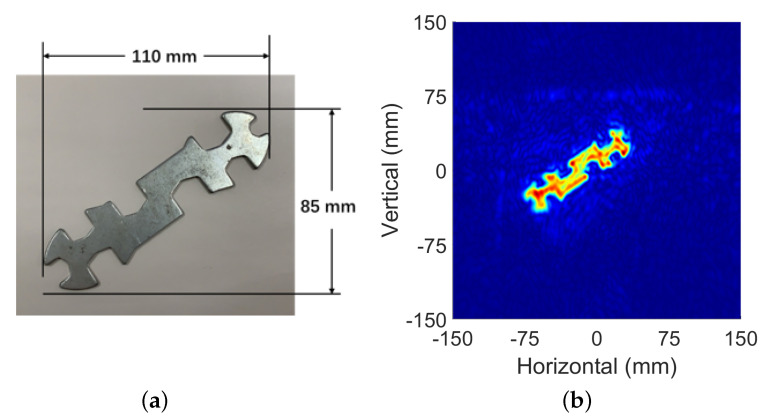
Image target: (**a**) optical image; (**b**) reconstruction results of the SAR data where $d_{x}$ = 0.91 $d_{y}$ = 1 mm.

**Figure 12 sensors-22-05548-f012:**
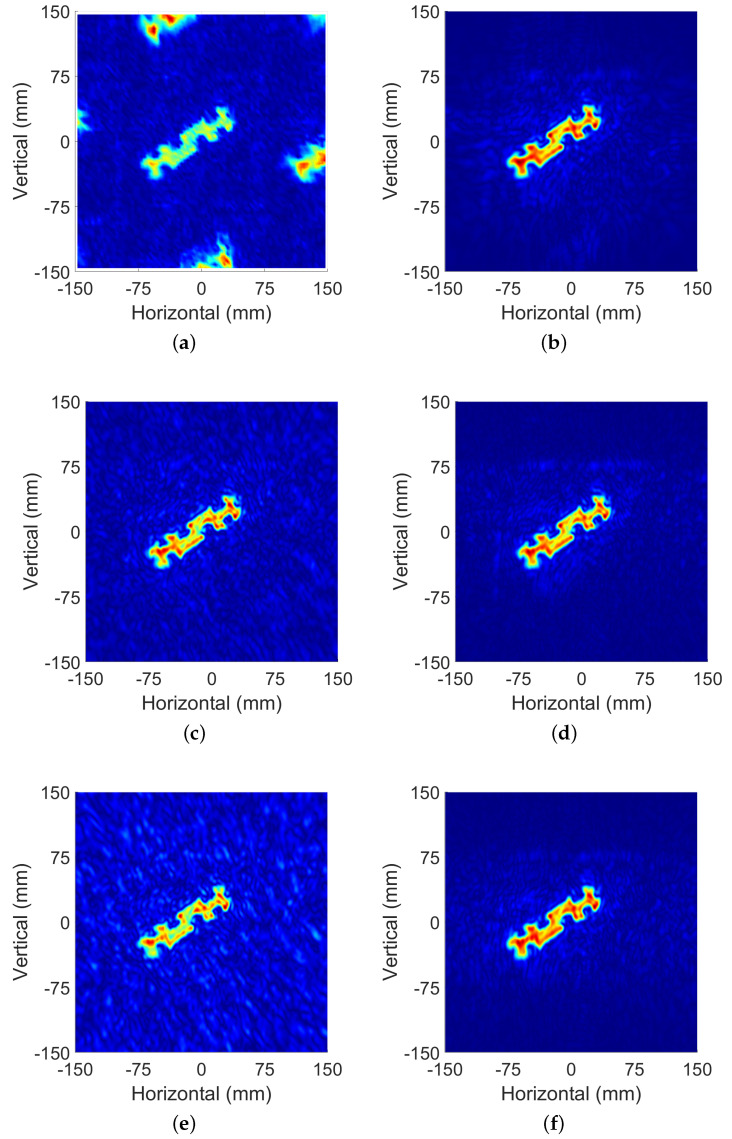
Reconstructed images under different sampling conditions: (**a**) image reconstructed from under-sampled data with a $d_{x}$ = 3.84 $d_{y}$ = 4 mm scan step; (**b**) image reconstructed from under-sampled data with a $d_{x}$ = 3.84 $d_{y}$ = 4 mm scan step with proposed method. (**c**) Reconstruction results of the SAR data with USR = 60% by match filter method. (**d**) Reconstruction results of the SAR data with USR = 60% by proposed method. (**e**) Reconstruction results of the SAR data with USR = 30% by match filter method. (**f**) Reconstruction results of the SAR data with USR = 30% by proposed method.

**Table 1 sensors-22-05548-t001:** Simulation parameters of near field synthetic aperture radar imaging.

Scan Parameters	$Z_{0}$	300 mm
	$D_{x}$	200 mm
	$D_{y}$	200 mm
	Step	1 mm
Radar Parameters	$f_{0}$	77 GHz
	Frequency slope	$6.314\times 10^{13}$ HZ/s
	Bandwidth	4 GHz

**Table 2 sensors-22-05548-t002:** Error of recovered data with different under-sampled rate (USR).

USR	10%	30%	50%	70%	90%
Error in this paper	54.34%	8.7%	5.34%	1.14%	0.43%
Error in ref. [22]	68.05%	10.01%	6.34%	2.3%	1.61%
Error in ref. [23]	73%	6.5%	5.9%	5.4%	4.7%

**Table 3 sensors-22-05548-t003:** *NMSE, SSIM, PSNR* of reconstructed image in different conditions.

Large scan interval	Step	2 mm	3 mm	4 mm	5 mm	6 mm
	NMSE	0.54%	1.62%	4.16%	7.22%	8.16%
	SSIM	89.09%	70.33%	53.66%	39.07%	32.75%
	PSNR	216.95	212.14	205.13	204.87	203.67
Random sparse data	USR	80%	60%	40%	30%	20%
	NMSE	0.48%	1.42%	3.87%	6.38%	10.10%
	SSIM	93.07%	76.51%	57.99%	48.78%	42.63%
	PSNR	217.44	212.72	208.3676	206.2001	204.2016

**Table 4 sensors-22-05548-t004:** *NMSE, SSIM, PSNR* of reconstructed image in different conditions.

Large scan interval	Step	2 ×	3 ×	4 ×	5 ×	6 ×
	NMSE	2.68%	3.44%	5.73%	8.34%	13.06%
	SSIM	80.22%	64.76%	47.76%	36.70%	26.6%
	PSNR	169.34	168.51	165.09	164.38	163.94
Random sparse data	USR	80%	60%	40%	30%	20%
	NMSE	0.69%	2.11%	5.16%	8.01%	12.47%
	SSIM	98.07%	90.2%	69.53%	52.08%	33.97%
	PSNR	176.21	171.82	168.14	166.04	164.24

## Data Availability

The data is not applicable due to privacy and ethical restrictions.

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
