# Peer review of "Near-Field High-Resolution SAR Imaging with Sparse Sampling Interval"

_sensors, 2022, doi:10.3390/s22155548_

Round 1

Reviewer 1 Report

This paper established a novel near-field mmWave SAR imaging model in the case of large space interval and sparse sampling.The amount of data in the imaging process can be greatly reduced. Some suggestions and questions are as follows:

1."SISO" should be given the full name in this paper, just like "MIMO", "FMCW", although it is explained at the end of the paper.

2.There are many kinds of under-sampled recovery reconstruction methods, it is suggested that supplement the comparative experiments, to clarify the advantages of the proposed algorithm.

3.It is suggested that analyse the time complexity of the proposed algorithm.

4.It is suggested to quote more recent papers.

Author Response

Anonymous comments for transmission to the AUTHOR(S):

1."SISO" should be given the full name in this paper, just like "MIMO", "FMCW", although it is explained at the end of the paper.

We have revised and improved the usage of abbreviations. The full name is added where the abbreviations first appear in the text to avoid confusion.

2.There are many kinds of under-sampled recovery reconstruction methods, it is suggested that supplement the comparative experiments, to clarify the advantages of the proposed algorithm.

Compared with CS method, this method performs better in error under the same usr data conditions. In addition, this method can realize reconstruction without designing the observation matrix.

3.It is suggested that analyse the time complexity of the proposed algorithm.

The time complexity has not been analyzed temporarily, because the current method adopts the method of traversing points. This is also a direction we will optimize. However, we have counted the execution time of the algorithm. When usr=30%, the time of image restoration and reconstruction by the method proposed in this paper is about 0.27s, which is about twice the reconstruction time of directly using the matched filter algorithm (0.127s).

4.It is suggested to quote more recent papers.

More relevant literatures have been cited.

Reviewer 2 Report

In this paper, an effective Near-Field SAR Imaging method with large sampling interval or under-sampling condition is proposed. In general, the authors’ method cannot be fully validated by current simulations. The authors should further enhance their experiments.

1. In abstract, ‘SIMO’ should be first given full name before abbreviation.

2. In simulations, the authors should conduct comparisons between their method with compressive sensing method. With this operation, the advantages of authors’ method can be highlighted.

3. In simulations, the experiment with single case, i.e. 30% under-sample data, is not sufficient to validate the authors’ method. The authors should discuss the performance of their method and compressive sensing method versus with different under-sample data.

Author Response

  1. In abstract, ‘SIMO’ should be first given full name before abbreviation.

We have revised and improved the usage of abbreviations. The full name is added where the abbreviations first appear in the text to avoid confusion.

  1. In simulations, the authors should conduct comparisons between their method with compressive sensing method. With this operation, the advantages of authors’ method can be highlighted.

The comparison of adding CS method is given in the simulation results. Compared with CS method, this method performs better in error under the same USR data conditions. In addition, this method can realize reconstruction without designing the observation matrix.

  1. In simulations, the experiment with single case, i.e. 30% under-sample data, is not sufficient to validate the authors’ method. The authors should discuss the performance of their method and compressive sensing method versus with different under-sample data.

For the cases of uniform large interval sampling and random sparse sampling, this method can effectively restore and reconstruct. The image results with usr=60% are added in this paper. Table 4 also gives a summary of more data. Gif in the bellow is the image result under different sampling conditions.

(GIFs can be found in attachment!)

Reviewer 3 Report

Reviewer

MDPI – Sensors

Manuscript Number: ID 1794941

Title: Near-Field High-Resolution SAR Imaging with Sparse Sampling Interval

As requested, I have reviewed the above-titled paper for potential publication in the Sensors - MDPI Journal. I divided my comments in the sections presented as follows.

Contribution

This paper proposes to examine near-field mmWave SAR imaging with respect to formulate a new model to better address large space interval and sparse sampling. The authors mention that they are proposing a novel approach in this area by means of using 1D-FFT applied to the obtained time domain signal. The proposal intends to obtain full aperture data by filling the empty sampling points with amplitude and phase compensation based on the stronger correlation expected of adjacent sampling points.

To evaluate the feasibility of this proposal, a near-field mmWave imaging prototype is built to validate the novel method. Statistical metrics, namely NMSE (normalized root-mean-square error), PSNR (peak signal-to-noise ratio)  and SSIM (structural similarity), were used to evaluate the performance of the method under different scenarios.

The authors claim that their approach is different from the compressive sensing method (e.g.,reference  number 20), since their method uses the range focusing data as basis for reconstruction and does not need to design the observation matrix. In addition,  the authors state (Lines 67-68):”The full aperture sampling data matrix is reconstructed from the under-sampling  data inconsistent with the Nyquist theorem.”

I found the manuscript has an interesting goal to be pursued and presents the conditions to be published in the MDPI – Sensors Journal. However, the text should be fully revised.

Therefore, the more specific comments and questions with respect to the manuscript paper are going to presented jointly with the evaluation of the contribution of the manuscript in the next section. They are provided below with more details with respect to the methodological approach. I think it would be interesting to have some feedback from the authors and also to have a revised version of the manuscript in order to better refer to the points I will raise in the next paragraphs. That might also lead to explore or reflect about different scenarios still not well and thoroughly explored by the authors in the proposed paper but that deserves attention.

Please, see further comments in the attached file. 

Author Response

  1. Methodological approaches are neither fully nor adequately reported. There are many gaps along the text. The workflow of the whole methodological approach is not presented. Figure 4 is a trial to present the flow diagram for the proposed method. It comes in Section 2 -Module and algorithm. However, Figure 4 does not represent the full methodological approach employed in this research work. Therefore, a special Figure should be previously presented to the reader encompassing a full framework designed for the manuscript. More than that, in the last paragraph of the introductory Section, the authors say: " Section II reviews the SAR models based on FMCW signal and the imaging algorithm based on amplitude and phase compensation recover. This section 2 should be split in two parts. Section 2a (or simply 2) should refer to Background on SAR models on FMCW signal, while Section 2b (or simply 3) should refer to Methodology- module and algorithm.

The method proposed in this paper is an attempt and exploration based on the matched filter reconstruction method. Therefore, the first part of the second chapter mainly reviews the process of matched filter algorithm, and its completion process has been added.

  1. Please, also revise the abstract, since it seems to be incomplete in terms of the results achieved (statistical metrics). On the other hand, in certain points, the abstract seems to be quite particular and not more general. Furthermore, it seems to be about the same as the paragraph written for the section Conclusion, which is too short. Indeed, comments on results and conclusions are not clearly stated in the abstract. Concluding remarks (Section of Conclusion) is also too short. There is much more to say to better wrap-up the manuscript, notably with respect to the results and comparisons presented. It is missing a discussion section, before the so-called section of Conclusion, to address all the pros and cons and limitations of the proposed method.

At the same time, the abstract and conclusion have been greatly modified and improved. The main feature of this method is that it allows the under sampling process in MF reconstruction, and will not be affected by the correlation phenomenon caused by the high side lobe of under sampling. The simulation and experimental results clearly show that this method can effectively reconstruct the target in both undersampling cases, and effectively suppress some adverse phenomena, especially ghosting. Unlike CS method, this method does not need to design observation matrix. The test platform of this paper also confirms that this method can recover the range focused data directly (this process can be realized by on-board DSP) without sampling the time-domain data, which greatly reduces the cost of data transmission.

  1. On the other side, I think it would be interesting to have some feedback from the authors and also to have a discussion with respect to the authors call as "novel method". The authors refer to the use of SAR images. In particular, I have not seen a discussion with respect to the oscillation of the electric field (HH, VV, HV, VH) with respect to the scene. Is that an issue to be taken into account in the proposed algorithm?

Thank you for your reminder and suggestion, which is very important for our next research. At present, this paper does not discuss electric field, but I find it a very interesting field. Under different antenna polarization modes, substances with different characteristics can be screened and filtered. In the next step, we will try to use different polarization methods for substances with different characteristics.

  1. In addition, the authors mention a lD-FFT (time domain and frequency domain). It does not seem to have any novel approach in such procedure. What about 2D and 3D approaches (e.g. references cited by the authors)? Could the authors be more clear on that and better situate the proposal under the state-of-the art in this area? Could the authors better explain this statement: (Lines 67-68):"The full aperture sampling data matrix is reconstructed from the under-sampling data inconsistent with the Nyquist theorem."

In near-field SAR, spatial sampling generally needs to meet a certain spacing, which can generally be considered as λ/ 4. This can be explained in detail in quotation 22 of this article. First, the conventional CS and MC methods recover the time-domain data during recovery. First, the time-domain sampled signal is completed, and then the subsequent image reconstruction process is carried out. In this paper, for the data sampled by the radar at some spatial points, range focusing will be carried out first, and a spatially incomplete 2D data will be obtained. Then, amplitude and phase compensation will be calculated through micro range change to fill the focused data. Then it is reconstructed directly through MF method, which greatly reduces the pressure in the process of data transmission. The process of range focusing can be realized in the radar module on-board DSP we use. This means that the volume of data that needs to be transferred to PC will be greatly reduced.

Therefore, when the complete spatial sampling point is 100*100, the method in this paper can try to recover the original 100*100 sampling point data through a small number of sampling points, and then reconstruct the image to obtain a reconstructed image close to the full sampling condition.

  1. It is necessary to be more clear on the original contribution of the manuscript. The authors show some results and evidences of fair performances of the proposed algorithm for SAR image formation/reconstruction (mmWave SAR) based on sampling in the time domain and frequency domain. I believe the authors could significantly improve the presentation of their results in the manuscript in order to better convince readers with respect to their original contribution. Maybe some more details for some of the figures and tables could have been added to improve illustration of the results.

In addition, could the authors further and better elaborate results using also the spatial domain to better visualize the performance under both domains ( e.g. Figures 6, 7, 10 and 11? Do the authors think that those statistical metrics as proposed are enough to evaluate the performance of the algorithm? Which would be the challenges to provide a more complete framework of analysis in terms of the implementation of the proposed method?

This paper is based on the lack of spatial sampling points, first restore the focused data, and then reconstruct the image. Therefore, we pay more attention to the similarity of reconstruction results based on the same method (MF method in this paper). This can well reflect the restoration performance of this method to the imaging scene. In Tables 2 and 4 of the article, NMSE, SSIM and PSNR under simulation and actual test conditions are given to compare the image similarity. I think this is more important for us. In the following GIF, we give the reconstruction results from usr=20%~90%. In addition, the reconstruction results at different intervals under the condition of uniform sampling are also shown in the figure below.

(Gifs can be found in attachment!)

  1. I would like to hear from the authors a more complete evaluation of the proposed algorithm with respect to different targets (e.g., land use and land cover, topography, roughness, soil moisture, vegetation, ocean and ice environments). How the method could contribute with respect to those range of applications? It seems that this is alsoan issue as we talk about how to optimize the formation and reconstruction of SAR images. Parameterization in this respect is quite important to better produce SAR images.

More than that, how the proposed algorithm addresses such topics? Which are the limitations of the proposed algorithm? The authors do not discuss such relevant issues in contrast to the literature (see, for example, Ulaby et al. 1981). Is there any comment on interferometry and altimetry, for example? What about to signal-to-noise ratio and backscattering coefficient - sigma nought? What about speckle in such type of mm Wave SAR images?

At present, this paper only considers the near-field imaging of metal objects. In our recent experiments, we also increased the detection of hidden objects, and the effect is still acceptable. At present, our research mainly focuses on the optimization of improving imaging accuracy and reducing the amount of data, and we have not carried out research on other different target areas. As for the imaging scenes such as vegetation and soil, we have not covered them for the time being, and we will try them in our recent work.

  1. Which are the next steps?

Next, we will optimize this method. Because the current method is traversal to recover, the execution efficiency is low. At the same time, we look forward to its application in 3D imaging.

  1. As a final comment, in addition to the previous comments and questions raised, please also revise the text.

We have revised and improved the usage of abbreviations. The full name is added where the abbreviations first appear in the text to avoid confusion. Illustrations and tables have also been modified to some extent. Appropriate instructions are added to the corresponding paragraphs. Finally, the grammatical errors and the use of words in the article have also been improved and changed.

Reviewer 4 Report

The authors have proposed a near-field SAR imaging method with large spatial sampling intervals (under-sampling conditions). From the experimental and real data points of view, the paper is interesting. However, the paper should be double-checked on the style of writing and use of the English language.

For example, in lines 2-4,

"In this paper, a novel near-field mmWave SAR imaging model is established, in the case of large space interval and sparse sampling, 1D-FFT is performed on the obtained time domain signal and focus to obtain the slice data of the estimated distance"

- It is too long and should be decomposed into shorter sentences.

- "and focus to"  is not correct

- repetitive use of "obtain "-->obtained time domain signal ...to obtain

- time-domain

Line 9, SISO, acronyms should be defined just together with their first use

Line 11, what is the traditional method?

line 14, (dot) is required at the end.

Line 20-22: a very short paragraph is not recommended.

Line 48-49: check this sentence for the grammar "allows obtain and reconstruct sparse or compressible signals for image construction."

Line 52: a CS-based method

Line 64: why capital letters in Near Field? in contrast to the line 70

Page 3, between (1) and (2)==> bring (and Not Bring )

Page 4, between (9) and (10) ==>where (and Not Where)

Page 5, place a name for the algorithm instead of the Algorithm caption

Anyway, I believe that the manuscript should be read another time to obviate its language problem. In the current form, it is not easy to follow some sentences.

in the last paragraph of the Introduction, the contribution of this work should be clearly highlighted.

In the end, the proposed method should be compared with the state-of-the-art methods, in particular, recent publications in the past five years. 

Author Response

  1. Anyway, I believe that the manuscript should be read another time to obviate its language problem. In the current form, it is not easy to follow some sentences.

We have revised and improved the usage of abbreviations. The full name is added where the abbreviations first appear in the text to avoid confusion. We have modified some sentences of the article to make it easier to understand and read.

  1. In the last paragraph of the Introduction, the contribution of this work should be clearly highlighted.

The comparison of adding CS method is given in the simulation results. Compared with CS method, this method performs better in error under the same USR data conditions. In addition, this method can realize reconstruction without designing the observation matrix.

  1. In the end, the proposed method should be compared with the state-of-the-art methods, in particular, recent publications in the past five years.

Based on this method (Spatial under-sampling image reconstruction method based on amplitude and phase compensation), we designed a highly practical two-dimensional imaging test-bed, and reconstructed the image under two kinds of under sampling conditions. Through various parameters, we evaluated the difference between the results and the full sampling conditions. When usr=30%, the error is 5.74%. The following figure shows the imaging results of this method under different sampling conditions.

(GIFs can be found in attachment!)

Round 2

Reviewer 1 Report

The comments have been replied accordingly, there are no other suggestions. 

Author Response

Double-checked on the style of writing and use of the English language has been done.

Reviewer 2 Report

The paper has been improved after revision.

Author Response

(The authors gave the same response as above.)

Reviewer 3 Report

Reviewer

MDPI – Sensors

Manuscript Number: ID 1794941-V2

Title: Near-Field High-Resolution SAR Imaging with Sparse Sampling Interval

As requested, I have reviewed the revised version of the above-titled paper for potential publication in the Sensors - MDPI Journal. I divided my comments in the sections presented as follows.

Contribution

This paper proposes to examine near-field mmWave SAR imaging with respect to formulate a new model to better address large space interval and sparse sampling. The authors mention that they are proposing a novel approach in this area by means of using 1D-FFT applied to the obtained time domain signal. The proposal intends to obtain full aperture data by filling the empty sampling points with amplitude and phase compensation based on the stronger correlation expected of adjacent sampling points.

To evaluate the feasibility of this proposal, a near-field mmWave imaging prototype is built to validate the novel method. Statistical metrics, namely NMSE (normalized root-mean-square error), PSNR (peak signal-to-noise ratio)  and SSIM (structural similarity), were used to evaluate the performance of the method under different scenarios.

The authors claim that their approach is different from the compressive sensing method (e.g.,reference  number 20), since their method uses the range focusing data as basis for reconstruction and does not need to design the observation matrix.

I found the manuscript has an interesting goal to be pursued and presents the conditions to be published in the MDPI – Sensors Journal. The text has been revised accordingly previous suggestions, however there are still some mistypes and  minor  editing corrections to be done  in the text.

In addition, the authors presented fair reflections with respect to some comments made regarding the first version of the manuscript and were able to improve the manuscript with some adjustments that have been made in the text.

Please, see further comments in the attached file.  

Author Response

There are still some minor corrections to be revised by the authors for the final version of the manuscript, including legends for Figures and Tables.

We have revised and corrected all language and grammatical errors, all marked red in the article. We have revised the legends of the figures and tables you mentioned in the review.

Here are some changes in the content:

Line 100 - withdraw one dot ; revise the sentence - marked in red- it is not well

written- use " ... data. A matched filter ... then multiplied by the range compressed

data slice represents a specific distance."; in addition, withdraw a blank space

before the fmal dot

After that, a matched filter for a specific distance is generated to calculate Hadamard product with the azimuth compress.

Line 222 - please revise" 300 mm and 300 mm"; are the authors referring to the

region "300 m X 300 mm"?;

Figure 11 shows the target image within a visible region of 300 ×300 mm2.

Line 242 - 243 - revise and rewrite the sentence - use "Table 4 presents a

summary for the different imaging data results retrieved based on different undersampled

rates (USRs); please double check to verify what the authors would like to

really express here-in;

The images reconstructed by our proposed method with different under-sampling conditions are evaluated by comparing with the result reconstructed from full sampled data.

Reviewer 4 Report

First, the point-by-point reply letter is not complete, I mean the first half of the items, which I had written in the first round of peer-reviewing, have not been mentioned in the authors' reply letter. The authors have revised the paper based on some of the comments, however, it is expected to see their replies to all the items in the response letter. Moreover, unfortunately, I did not find the authors' replies to the remaining items convincing. For example, an extensive comparison with previous methods from the last five years was asked, but the authors' reply does not exactly refer to the point.  

Author Response

First, the point-by-point reply letter is not complete, I mean the first half of the items, which I had written in the first round of peer-reviewing, have not been mentioned in the authors' reply letter. The authors have revised the paper based on some of the comments, however, it is expected to see their replies to all the items in the response letter.

Double-checked on the style of writing and use of the English language has been done. We have revised and corrected all the errors mentioned in the comment, and marked them in red.

Moreover, unfortunately, I did not find the authors' replies to the remaining items convincing. For example, an extensive comparison with previous methods from the last five years was asked, but the authors' reply does not exactly refer to the point. 

How to effectively reduce the size of sampling data and ensure the imaging quality is a challenging problem. In the introduction, we combed the literature in recent years. Compared with MIMO method, this method does not need multi-channel phase calibration, and can effectively reduce the number of sampling points in the process of radar scanning. Compared with CS method, the method proposed in this paper does not need to design the observation matrix, and can effectively recover the data under the condition of sparse sampling. At the same time, in the process of simulation, we compare this method with CS method. Table 2 also shows the method proposed in this paper in detail and compares it with the data in recent articles.

This manuscript is a resubmission of an earlier submission. The following is a list of the peer review reports and author responses from that submission.